# Effects of Maize Straw Incorporation on Soil Water-Soluble Organic Carbon Fluorescence Characteristics

**DOI:** 10.3390/plants15010004

**Published:** 2025-12-19

**Authors:** Enjun Kuang, Jiuming Zhang, Gilles Colinet, Ping Zhu, Baoguo Zhu, Lei Sun, Xiaoyu Hao, Yingxue Zhu, Jiahui Yuan, Lin Liu, Jinghong Ji

**Affiliations:** 1Key Laboratory of Soil Environment and Plant Nutrition of Heilongjiang Province, Heilongjiang Academy of Black Soil Conservation and Utilization, Harbin 150086, China; kuangenjun2002@163.com (E.K.);; 2Department of Biosystems Engineering (BIOSE), Gembloux Agro-Bio Tech of Université de Liège, 5030 Gembloux, Belgium; 3Jilin Academy of Agricultural Sciences, Changchun 130119, China; 4Jiamusi Branch, Heilongjiang Academy of Agricultural Sciences, Jiamusi 154007, China; zhubaoguo82@163.com; 5College of Agriculture, Heilongjiang Bayi Agricultural University, Daqing 163319, China

**Keywords:** maize straw incorporation, water-soluble organic carbon, three-dimensional fluorescence spectra, PARAFAC analysis

## Abstract

Farmland soil water-soluble organic carbon (WSOC), serving as a labile carbon substrate for microbial utilization, demonstrates pronounced sensitivity to land-use modifications and agricultural management practices. This study systematically investigated the impacts of long-term straw incorporation frequencies—including annual (S-1), biennial (S-2), and triennial (S-3) return patterns—on WSOC distribution across 0–20 cm and 20–40 cm soil profiles. Through the integration of three-dimensional excitation–emission matrix (EEM) fluorescence spectroscopy with parallel factor analysis (PARAFAC), we elucidated structural characteristics and humification dynamics associated with different incorporation regimes. The results showed a depth-dependent WSOC distribution pattern with higher concentrations in surface soils (0–20 cm: 261.2–368.9 mg/kg) compared to subsurface layers (20-40 cm: 261.8–294 mg/kg). Straw incorporation significantly increased WSOC content in the 0–20 cm of 16.9%~21.7% and 20–40 cm soil layers of 6.2%~12.3%. Biennial return had the lowest WSOC/SOC ratio, indicating enhanced stability of the soil organic carbon pool. Spectral indices—including the fluorescence index (FI, 1.59~1.69), biological index (BIX, 0.90~0.95), and humification index (HIX, 0.64~0.74)—collectively indicated that WSOC predominantly consisted of microbially processed organic matter with a low degree of humification. PARAFAC modeling resolved two fluorescent components: C1 (humic acid-like substances, 47.4–50.4%), C2 (soluble microbial metabolites, 49.6–52.6%). This systematic investigation provides mechanistic insights into how straw management temporality regulates both quantity and quality of labile carbon pools in agricultural ecosystems.

## 1. Introduction

Incorporation of crop straw as a resource into agricultural fields improves the soil structure and nutrients [1,2], as straw is rich in C, N, P, K, and other trace elements. This practice has been widely adopted to improve soil fertility improvement [3,4]. The straw production in Northeast China is 156.45 million tons, accounting for 18.1% of the total straw resources in China in 2022 [5]. However, low temperatures and a nearly six-month freezing period in this region prevent a significant portion of straw from decomposing fully. Such incomplete decomposition leads to soil compaction and moisture loss, adversely affecting spring sowing quality in the following year and resulting in uneven seeding emergence and further drying. Guided by the principle of ‘prioritizing in-field utilization, supplemented by off-field utilization,’ comprehensive and circular straw utilization measures are being intensified [6]. Straw incorporation methods are broadly categorized as direct and indirect. Direct incorporation of maize straw mainly includes deep-layer mixing through rotary tillage, deep plowing, and deep turning techniques, as well as surface mulching after shredding [7]. When combined with deep plowing, straw incorporation can further enhance nutrient levels in deeper soil layers [8], thereby encouraging deeper root penetration of crops and increased crops yield [9,10]. Moreover, return straw to deep soil increases the sequestration of straw-derived carbon [9], promotes the accumulation of humus [10], and improves overall soil nutrients and fertility.

Water-soluble organic carbon (WSOC) is a vital, biologically active constituent of soil nutrient pool and plays a crucial part of material cycling [11]. It refers to the organic components in soil samples that are soluble in water at room temperature and natural pH conditions [12]. It can also be transformed into other components of soil organic carbon. Straw incorporation has been demonstrated to significantly increase soil WSOC content [13]. The soil organic carbon (SOC) content not only significantly affects the fluorescence peaks of WSOC but also interacts with tillage practices and straw incorporation frequency. These factors collectively lead to varying degrees of WSOC humification [14,15,16]. Research has shown that after straw incorporation, the content of fulvic acid in fluorescent organic carbon fraction increases, accompanied by simplification of its molecular structure [17]. The source, distribution, and compositional characteristics of WSOC can be characterized using three-dimensional excitation–emission matrix (EEM) spectroscopy combined with multivariate statistical analyses such as parallel factor analysis (PARAFAC) [18,19,20,21]. This analytical approach reveals distinct fluorescence signatures of WSOC components, which are strongly influenced by the soil organic carbon (SOC) content [18].

In the Danchi Lake area, structural characteristics of WSOC were investigated using UV-Vis and EEM techniques, revealing that fulvic acid-like substances dominate, with no clear indications of exogenous and endogenous origins [22]. To improve the utilization of maize straw in the fermentation of organic materials, a reference composting ration has been proposed based on the fluorescent components of dissolved organic matter (DOM) in mixtures of straw and cow dung [23]. Furthermore, by analyzing WSOC composition and the soil structure across different soil types, the adsorption behavior of WSOC has been clarified, providing guidance for rational compost application tailored to specific soils [24]. Therefore, this study employs the EEM-PARAFAC method to analyze the composition, characteristics, and distribution of WSOC under straw incorporation conditions.

We hypothesize the practice of maize straw application influences WSOC fractions of fluorescent materials. Herein, the main objectives of this study were to (1) determine whether annual straw return increases the instability of soil carbon pool and (2) annual straw return to the field will reduce the stable components of WSOC and decrease humification. This study aims to characterize changes in WSOC content and composition induced by straw incorporation.

## 2. Materials and Methods

### 2.1. Site Description

The study was conducted in an experimental station (43°30′ N, 124°49′ E) located in Gong Zhuling, Jinlin Province, China, a representative black soil region of Northeast China. Soil was a typical black soil (aligned with Phaeozem in the Chinese soil taxonomy) developed from loess parent material, in which particle size distribution was 38.3% for sand, 29.9% for silt, and 31.8% for clay [25]. Initiated in 2011, this five-year field experiment (2011–2016) focused on a monocropping maize system under temperate continental monsoon climate conditions. The site exhibits the following climatic characteristics: mean annual temperature is 5.2 °C, ≥10 °C active accumulated temperature from 2600 to 3000 °C. The annual sunshine duration ranges from 2500 to 2700 h and annual precipitation ranges from 450 to 650 mm (70% concentrated in June–September). Initial baseline soil characterization (0–20 cm depth) is as shown in Table 1.

### 2.2. Experimental Design

The trial employed maize (*Zea mays* L.) straw as incorporation material, with the following experimental components: post-harvest maize straw (average yield: 10,000 kg·hm^−2^) was mechanically crushed (<5 cm fragments) by a John Deere S660 (320 HP) (John Deere, East Moline, IL, USA) combine harvest with an integrated shredding system and deep-tilled (30–35 cm) by a EurOpal-4 hydraulic flip plow (200 HP) (Shandong Nongxin Machinery Group Co., Ltd., Weifang, Shandong, China). Four treatments were established as annual incorporation (S-1), biennial (S-2), and triennial incorporation (S-3) patterns, and no straw incorporation (NS). Randomized complete block design with three replicates with individual plot size of 500 m^2^. The experimental plots were arranged using a random design with three replications, and each plot covered an area of 500 m^2^. Agronomic practices were employed equally to all the plots, such as fertilization, irrigation, and pest control. The annual fertilization amount included the pure nitrogen (N) 165 kg·hm^−2^, P_2_O_5_ 82 kg·hm^−2^, K_2_O 82 kg·hm^−2^.

### 2.3. Measurement Methods

Soil sampling was conducted in April 2016 following a standardized protocol: five points arranged in an S-pattern configuration were selected within each experimental plot. With stratified collection from two soil depths (0–20 cm and 20–40 cm). At each plot, five depth-specific subsamples were homogenized to form representative composite samples. Post-collection processing involved manual removal of residual straw fragments and lithic particles, followed by controlled air-drying at ambient laboratory conditions. The desiccated samples were sequentially sieved through 2 mm and 0.01 mm mesh screens using mechanical shakers for spectroscopic and chemical analyses.

Determination of soil organic carbon (SOC): the air-dried soil samples was weighed at 0.01xx g and subjected to acid digestion using a 2 mol·L^−1^ hydrochloric acid solution. Subsequently, the resulting solution was filtered through a 0.45 µm membrane, and the SOC content was determined using a total organic carbon analyzer (multi-N/C 3100, Analytik Jena, Jena, Germany) [26].

Determination of WSOC content: a 3 g portion of air-dried soil was mixed with 30 mL of ultrapure water. The mixture was horizontally oscillated at a speed of 200 r·min^−1^ for 24 h at room temperature. After centrifugation at 12,000 r·min^−1^ for 20 min, the supernatant was filtered through a 0.45 μm membrane before being analyzed using a total organic carbon analyzer (Multi N/C 3100, TOC instrument, Analytik Jena GmbH, Jena, Thuringia, Germany) to determine the concentration of DOC [27].

Fluorescence spectrum: a small aliquot of solution rom each treatment was diluted with ultrapure water to standardize the WSOC concentration to 10 mg·L^−1^, thereby minimizing the influence of WSOC concentration on the fluorescence measurements. The EEM fluorescence spectra were then acquired using a fluorescence spectrometer (Hitachi F-7000, Tokyo, Japan). There was a scanning range of 200 to 600 nm for both excitation (Ex) wavelength and emission (Em) wavelength, with a bandwidth of 10 nm and a scanning speed of 1 200 nm·min^−1^. Ultrapure water was used as the blank [23].

Fluorescence index (FI) was calculated as the ratio of emission intensity at 450 nm to that at 500 nm when the excitation wavelength was fixed at 370 nm [28]. This index is widely used to infer the origin of humic substances in WSOC. Typically, an FI value below 1.4 indicates that WSOC is predominantly derived from the decomposition of external plant materials, such as litter and root exudates. In contrast, an FI value exceeding 1.9 indicates that WSOC originates mainly from metabolism and degradation within the soil. FI values range between 1.4 and 1.9, suggesting a mixed contribution from both plant and microbial sources [29].

Biological index (BIX) was calculated as the ratio of emission intensity at 380 nm to that at 430 nm under an excitation wavelength of 310 nm [30]. Typically, a BIX value between 0.6 and 0.7 reflects a low contribution of autochthonous sources to the WSOC pool. Values in the range of 0.7–0.8 suggest a moderate presence of recently produced autochthonous components, whereas values between 0.8 and 1.0 indicate a strong contribution from recently derived autochthonous WSOC. A BIX value greater than 1.0 signifies that the organic matter is predominantly autochthonous and newly generated [29].

Humification index (HIX) was determined as the ratio of the integrated emission intensity to the sum of integrated intensities over 435–480 nm and 300–345 nm at an excitation wavelength of 254 nm. This index is used to evaluate the extent of WSOC humification, with higher values indicating a greater degree of humification [31].

PARAFAC analysis was conducted in MATLAB 2013a using the DOMFlour toolbox [32]. To enhance model accuracy and interpretability, Raman scattering effects were removed by first subtracting the blank signal and then applying interpolation across the Raman scattering regions.

### 2.4. Statistical Analysis

Statistical analysis was performed using Excel 2010 and SPSS 22.0. One-way analysis of variance (ANOVA) was applied for comparative analysis at a significance level of α = 0.05. The three-dimensional fluorescence spectra were visualized, and PARAFAC was conducted in MATLAB 2013. Fluorescence spectral indices were calculated through area integration using Origin 2021.

## 3. Results

### 3.1. Dynamics of WSOC and WSOC/SOC Ratios

As demonstrated in Figure 1, at the 0–20 cm soil layer, SOC content was similar under the S-1 and S-2 treatments, both exceeding 20 g·kg^−1^, while those under the S-3 and NS treatments were also comparable. In the 20–40 cm soil layer, the S-1 treatment showed the highest SOC content, but SOC displayed a decreasing trend with increasing frequencies of straw return, following the order S-1 > S-2 > S-3 > NS. Significant differences were observed among all treatments (*p* < 0.05).

WSOC content exhibited a progressive decline with increasing soil depth across all treatments. Straw incorporation significantly enhanced WSOC concentrations in both surface (0–20 cm) and subsurface (20–40 cm) layers compared to the NS control (*p* < 0.05), with S-1 showing superior efficacy. Specifically, S-1 increased WSOC by 41.3% (0–20 cm) and 10.4% (20–40 cm) compared with NS, followed by S-3 (21.7% and 6.2%) and S-2 (16.9% and 12.3%). Annual incorporation consistently increased the WSOC content in the soil. Although the triennial treatment involved lower straw input, it resulted in a significantly higher WSOC content in the surface layer (0–20 cm) compared to the biennial treatment. As shown in Figure 1, the WSOC/SOC ratio displayed clear stratification across soil depths. In the 0–20 cm layer, the S-1 and S-3 treatments showed higher ratios (26.8%, 23.0%) than the S-2 treatment (6.7%), compared with NS treatment. Conversely, in the subsurface layers (20–40 cm), the S-1 treatment had a lower WSOC/SOC ratio than other treatments. The WSOC/SOC ratio is widely used as an indicator of SOC lability and its responsiveness to agricultural management practices. A higher ratio reflects greater SOC activity. Overall, annual incorporation enhanced the activity of liable organic carbon, while biennial treatment contributed to enhanced stability of the soil carbon pool.

### 3.2. Three-Dimensional Fluorescence Spectra of WSOC

The three-dimensional fluorescence landscape has been systematically categorized into five characteristic zones in prior research [33], with Region I and II corresponding to tyrosine-like and tryptophan-like protein compounds, Region III characterizing fulvic acid-like substances, Region IV indicating soluble microbial byproducts, and Region V representing humic-like materials, as detailed in Table 2.

In this study, the three-dimensional fluorescence spectra of soil WSOC under different treatments exhibited similar fingerprint characteristics, with three distinct fluorescence peaks observed in Figure 2. Fulvic acid-like fluorescence peak (Peak A): Excitation/emission (Ex/Em) = 230–250 nm/405–440 nm across treatments, generated by organic matter with high fluorescence efficiency and low molecular weight. Humic acid-like fluorescence peak (Peak C): Ex/Em = 270 nm/425–435 nm across treatments, produced by structurally stable organic matter with high molecular weight [20]. Soluble microbial product fluorescence peak (Peak T): Ex/Em = 300 nm/340 nm across treatments.

The three-dimensional fluorescence spectra revealed that humic acid-like substances dominated the WSOC composition in the experimental soils, indicating complex structural characteristics and a high degree of humification. Quantitative analysis based on spectral region integration (Table 3) revealed the following abundance order of WSOC components: humic acid-like substances > fulvic acid-like substances > soluble microbial metabolites > tryptophan-like proteins > tyrosine-like proteins. Notably, the integrated fluorescence intensities of regions I, II, and V in the NS treatment were significantly lower than those in straw return treatments. Previous studies have suggested that a red shift (i.e., a longer excitation wavelength) of the fluorescence peak in a given region indicates an increase in the aromaticity and molecular weight of the corresponding fluorescent substance. When observed in humic acid-like substances, it reflects a higher degree of molecular condensation [33]. In this study, after straw incorporation treatments, humic acid-like substances exhibited a blue shift in emission wavelengths accompanied by soluble microbial metabolites product peaks, suggesting decreased aromaticity and molecular weight. Compared to the NS treatment, S-2 and S-3 treatments showed red shifts (2–15 nm) in the emission wavelengths of humic acid-like substances. Conversely, humic acid-like substances in S-1 treatment demonstrated blue shifts in emission wavelengths, indicating an increase in their aromaticity and molecular weight.

The percentage distribution of fluorescence region integrals is presented in Table 3. The proportion of tyrosine-like proteins decreased with increasing straw incorporation frequency, with the NS treatment being significantly higher than S-1 (*p* < 0.05). Conversely, the ratio of tryptophan-like proteins increased with more frequent straw addition. Biennial straw return resulted in the highest humic-like substances and lower soluble microbial metabolites, whereas the S-1 treatment was associated with lower humic-like substances and high soluble microbial metabolites.

### 3.3. PARAFAC-Derived Fluorescence Components

According to the Fluorescence Regional Integration (FRI) analysis, humic-like substances constituted the primary fluorescent components in the samples, representing 54.3%~57.3% of the total fluorescence intensity. Subsequently, parallel factor (PARAFAC) analysis was employed to further resolve their intrinsic composition, which aided in the identification of two distinct fluorescent components (Figure 3).

C1: Humic acid-like substances (Ex/Em: 250–280 nm/420–460 nm), characterized by a single excitation–emission peak in the ultraviolet region.

C2: Soluble microbial metabolites (Ex/Em: 300 nm/360 nm), displaying dual excitation maxima associated with carboxyl-rich proteins.

Comparison of maximum fluorescence (Fmax) values revealed that soluble microbial metabolites (C2) dominated the fluorescent pool (49.56–52.59%), followed by humic acid-like substances (C1, 47.41–50.44%), which are shown in Table 4. The proportion of C1 in S-1, S-2, and S-3 treatments increased 4.48%, 6.41%, and 4.72%, respectively, compared to NS treatment. And the proportion of C2 in S-1, S-2, and S-3 treatments declined by 4.03%, 5.78%, and 4.26% in all the straw return treatments. It indicated that straw incorporation treatments increased humic acid-like substances and declined the soluble microbial by-product-like substances. Furthermore, biennial treatment benefitted to accumulate humic acid like substances, a stable material.

### 3.4. Fluorescence Spectral Characteristics

The fluorescence indices revealed distinct compositional differences among treatments, as shown in Table 5. The fluorescence index (FI) ranged from 1.59 to 1.69, with the NS treatment exhibiting the highest value (1.69 ± 0.02) and S-2 the lowest (1.59 ± 0.02), showing statistically significant divergence (Duncan’s test, *p* < 0.05). In contrast, the biological index (BIX) consistently exceeded 0.90 across all treatments, with NS marginally higher than straw incorporated groups. Humification index (HIX) demonstrated treatment-specific stratification: S-1 exhibited the lowest humification degree (0.64 ± 0.05), while S-2 achieved the highest (0.74 ± 0.03), suggesting biennial incorporation enhances organic matter stabilization.

### 3.5. Correlation Coefficients

From Figure 4, WSOC had extremely significant positive correlation with WSOC /SOC, and a negative correlation with FI and HA. HA had significant positive correlation with SOC, WSOC, and C/N; HIX had significant and extremely significant negative correlation with FI and BIX; and FI had a significant negative correlation with C/N.

## 4. Discussion

### 4.1. Influence of Different Treatments on the WSOC

Soil WSOC primarily derives from plant residues, the input of exogenous organic matter, and microbial metabolites. Its content is highly sensitive to minor changes in the soil environment and shows a strong positive correlation with SOC [33]. WSOC and SOC can undergo decomposition, transformation, and synthesis, often maintaining a dynamic equilibrium in soil systems [34]. In this context, returning crop straw to the field has been demonstrated as an effective management practice for enhancing soil WSOC [35,36]. In this study, application of exogenous straw significantly enhanced WSOC levels across all soil depths (0–40 cm), consistent with previously reported effects of organic amendments [13,15,23]. Research showed that deep tillage combined with straw return can increase organic carbon in the 0–40 cm soil layer, with a greater increase compared to subsoiling, no-tillage, and conventional tillage [37], because it can enhance root distribution and root residue content in deeper soil layers, effectively promoting the cultivation of deep soil [38]. Regarding the practice of straw return combined with deep tillage, some studies suggest that straw incorporation induces a positive priming effect, which enhances the mineralization rate of organic carbon while increasing soil organic carbon and active organic carbon [39]. On the other hand, deep tillage thoroughly mixes straw with the soil, accelerating straw decomposition, promoting organic carbon accumulation, and strengthening soil carbon sequestration capacity [38,40]. In this experiment, all treatments were subjected to the same tillage practice, and we attributed the changes in organic carbon content to straw incorporation. WSOC content also decreased with reduced straw return frequency. The no straw return treatment showed the lowest rate of WSOC increase, whereas the 0–20 cm layer under annual straw incorporation had the highest WSOC content. This surface accumulation of WSOC was likely due to the accumulation of organic inputs. The WSOC/SOC ratio offers valuable insights into the bioavailability of organic carbon and the effectiveness of agricultural management practices [41]. A higher ratio reflects greater SOC activity. According to the results of WSOC/SOC ratio, annual incorporation enhanced the activity of liable organic carbon, while biennial treatment contributed to the enhanced stability of the soil carbon pool.

### 4.2. Influence of Different Treatments on WSOC Component Fluorescence Indexes

The FI, BIX, and HIX serve as effective indices for characterizing the structural composition of soil humus [19]. Specifically, the fluorescence index (FI) reflects the origin of WSOC in soil, whereas the biological index (BIX) functions as an indicator of the characteristics and bioavailability of its autogenous sources [42]. In the study, the FI values ranged from 1.59 to 1.69 across treatments, indicating a mixed plant–microbial origin of WSOC. The NS treatment showed the highest mean FI value, which differed significantly from those under straw incorporation. In contrast, the biennial return (S-2) resulted in the lowest FI, suggesting stronger plant-derived characteristics in WSOC sources. BIX values exceeded 0.9 in all treatments, reflecting pronounced autochthonous microbial contributions to WSOC, regardless of straw management practices. Regarding humification assessment, the humification index (HIX) ranged from 0.64 to 0.74, indicating a relatively low degree of humification. This pattern implies a predominance of recently synthesized organic compounds derived from microbial and biological decomposition [43,44]. Among the treatments, S-1 showed the lowest HIX, while S-2 attained the highest values, implying that sustained straw incorporation promotes the accumulation of organic matter despite a reduced degree of humification, consistent with the presentation effect of WSOC /SOC mentioned earlier. These findings align with previous reports emphasizing the temporal amplification of humification differences with prolonged straw incorporation [45].

### 4.3. The Influence of Different Treatments on the WSOC Components

The fluorescence region integral (FRI) method has been widely applied to analyze the three-dimensional fluorescence spectra of dissolved organic matter in water and soil systems [46,47]. The method divides the excitation–emission matrix into five regions, representing distinct types of organic components: aromatic protein I, aromatic protein II, fulvic acid-like substances, soluble microbial metabolites, and humic acid-like substances. The integral volume of each region is calculated by quantifying the cumulative fluorescence intensity of compounds with similar properties, which is then standardized to obtain the fluorescence response proportional to the relative content of each component. In our experiment, annual straw return led to an increase in tryptophan-like protein substances and soluble microbial metabolites, a decrease in tyrosine-like proteins and humus-like substances, and no clear trend in fulvic acid-like substances. Studies have shown that protein-like components significantly influence the compositional dynamics of dissolved organic matter. A shift in the ratio of protein-like to humic-like fluorescence components was observed following straw return, indicating changes in the qualitative nature of soil-dissolved organic matter. Specifically, the annual straw return resulted in the lowest humification degree of WSOC among all treatments, with no significant difference observed in other regimes. The origin of dissolved organic matter can be attributed to both autochthonous sources (e.g., microbial and algal activity) and allochthonous inputs (e.g., humic materials) [47]. The application of organic amendments such as straw has been shown to enhance WSOC content, stimulate microbial decomposition and metabolism, and simplify the structural composition of fulvic acid components [48]. Senesi et al. [49] suggested that a blue shift (i.e., a shift toward shorter wavelengths) in the fluorescence peak indicates a decrease in the degree of humification, whereas a red shift (toward longer wavelengths) reflects an increase in humification. In this trail, the S-2 and S-3 treatments exhibited red shifts in humic acid-like substances, implying an enhanced degree of humification. In contrast, the S-1 treatment showed a blue shift. These observations are consistent with the HIX derived from fluorescence measurements: the S-1 treatment had the lowest HIX value, while the S-2 treatment showed the highest.

Parallel factor analysis (PARAFAC) can decompose complex three-dimensional fluorescence excitation–emission matrices into independent fluorescent components, thereby addressing the issue of interference and overlap between fluorescence spectra of structurally similar substances [50]. PARAFAC analysis was employed to further resolve their intrinsic composition, which aided in the identification of two distinct fluorescent components: soluble microbial metabolites and humic acid-like substances. Among these, soluble microbial metabolites dominated the fluorescent pool, followed by humic acid-like substances.

The C1 component, with its peak located within the Ex/Em range of 250–280 nm/420–460 nm, represents a typical Peak A [51]. It is categorized as a terrestrial, high-molecular-weight humic-like material. The C2 component exhibits a peak at Ex/Em = 300 nm/360 nm. Compared to the classical tryptophan-like position (Ex/Em = 275 nm/350 nm), its excitation and emission wavelengths are both longer, indicating that it contains more conjugated structures or aromatic rings than typical soluble proteins, along with stronger hydrophobicity and higher molecular weight. Based on the fluorescence indices analyzed above, the FI value reflects the combined influence of allochthonous and autochthonous sources, showing a tendency toward microbial-derived metabolites. Meanwhile, the relatively low HIX value suggests a lower degree of humification. Therefore, it is more appropriate to interpret the C2 component as an organic substance at a “moderate” stage of humification, primarily produced by microbial activity. It is more stable than proteins yet “fresher” than aged humic matter.

Soluble microbial products are direct byproducts of microbial activity, encompassing a range of organic substances secreted during metabolic processes. A higher concentration of SM indicates a more active microbial metabolic process. Straw incorporation generally increased the proportion of humic acid-like substances—a stable organic fraction—though this effect varied with the frequency of straw return. Specifically, the biennial straw return treatment was most conducive to the accumulation of humic acid-like substances. In contrast, the non-straw treatment exhibited higher levels of soluble microbial metabolites compared to straw incorporation treatments. This pattern may be attributed to limited organic inputs and carbon source deficiency under NS conditions. These findings contrast with previous reports where organic fertilization enhanced microbial activity and metabolite turnover [48,52]. In support of our interpretation, Corvasce et al. [53] suggested that reduced microbial metabolism can result in the accumulation of tyrosine-like and tryptophan-like proteins in soil. The strong adsorption affinity of soil matrices for these proteinaceous materials further reduces their microbial availability, thereby promoting their accumulation in the water-soluble organic carbon pool. In the natural state, the mineralization and decomposition of organic matter are accelerated, and organic–inorganic complexes are formed, leading to significant changes in the concentration, structure, and composition of SOC [54]. Following straw incorporation, the mineralization and decomposition patterns of SOM are altered, the small molecular fractions provide carbon sources for various microorganisms, enhancing microbial activity [55,56]. Additionally, the decomposition of straw requires extra energy, which is one of the reasons for the consumption of microbial metabolites. Studies have demonstrated that combined application of organic and inorganic fertilizers in black soil, red soil, paddy soil and fluvio-aquic soil can increase both WSOC and SOC contents [57], though differences exist among soil types. Red soil exhibits a higher proportion of microbial-derived components [58]. Black soil shows varying increases in amine compounds [59], and the combined application of chemical fertilizers with straw return in paddy soil leads to more complex and stable structures of aromatic compounds [60]. The use of different fertilization materials, straw return practices, and tillage practices contributes to an increase in the relatively simple-structured fulvic acid fraction within WSOC while simplifying its overall molecular structure [20,52].

### 4.4. Comprehensive Utilization of Straw

We also evaluated the carbon sequestration efficiency under different treatments. It was assumed that varying straw return frequencies corresponded to different amounts of straw incorporated into the soil. As shown in Table 6, the S-2 treatment exhibited higher carbon sequestration rates and efficiency compared to the other treatments. In contrast, the S-3 treatment, which involved a lower straw input, showed reduced carbon sequestration rates and efficiency. This outcome does not align with the expected beneficial effect of straw incorporation on soil carbon sequestration.

The total quantity of straw incorporated was comparable across treatments, but the key distinction lay in the frequency of return—specifically, whether straw was applied annually or in alternating years. This study captures the differential effects of consecutive versus intermittent straw addition. Previous long-term research on straw return rates has shown, for example, that applications of 400 kg ha^−1^ and 800 kg ha^−1^ decomposed completely within two years, while higher rates resulted only in partial decomposition. According to Meng et al. (2019), returning 75% of straw significantly enhanced soil nutrients, though higher return rates did not proportionally increase yield, likely due to reduced crop emergence and spike formation under excessive straw load under equivalent fertilization [61]. In this experiment, straw was annually, biennially, and triennially returned to the field; for example, straw biennial treatment was saved labor, materials, and cost, compared to straw annual half-rate return. The annual straw return introduces fresh carbon annually, accelerating the mineralization of plant residues. Within one year of incorporation, over two-thirds of straw organic matter is released as CO_2_ and water, leaving less than one-third for humification. As a result, easily decomposable compounds are rapidly metabolized, while recalcitrant components accumulate [62]. Su et al. (2020) reported that straw return raised both soil organic carbon (SOC) and CO_2_-C levels [63], whereas Naser et al. (2007) observed reduced CO_2_ emissions—a discrepancy suggesting that more frequent return and higher straw inputs generally enhance CO_2_ release [64].

In Heilongjiang Province, straw is used in five main categories—fertilizer, feed, fuel, substrate, and raw material—collectively termed the “Five Modernizations.” With government subsidies, the direct return rate has reached 95.0%. However, sustainable implementation still depends on subsidies, climatic conditions, mechanization quality, side effects of incorporation, and operational costs. Establishing a sound technical framework for straw return will help capitalize on this valuable resource, support waste-to-wealth initiatives, and advance the circular economy in agriculture—underscoring the need for more rational straw management models.

## 5. Conclusions

WSOC content decreased with soil depth, while straw application significantly enhanced WSOC concentrations in both the 0–20 cm and 20–40 cm soil layers. Notably, the annual straw incorporation displayed the highest WSOC/SOC ratio in the surface soil layer (0–20 cm) and biennial return led to a higher humification degree. In this study, straw incorporation generally increased the proportion of humic acid-like substances and reduced that of soluble microbial metabolites. This pattern implies that biennial straw return promotes the formation of macromolecular compounds and facilitates the development of stable humus.

## Figures and Tables

**Figure 1 plants-15-00004-f001:**
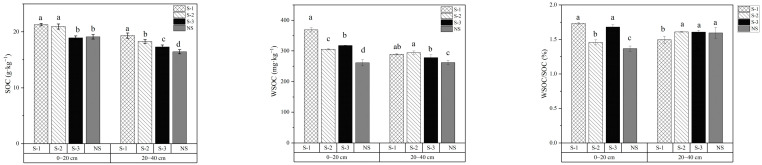
SOC, WSOC content, and WSOC/SOC ratios under different treatments in 0–40 cm soil profiles. Lowercase letters denote significant differences (Duncan’s multiple range test, *p* < 0.05) among treatments within the same soil layer. Error bars represent standard deviations (*n* = 3).

**Figure 2 plants-15-00004-f002:**
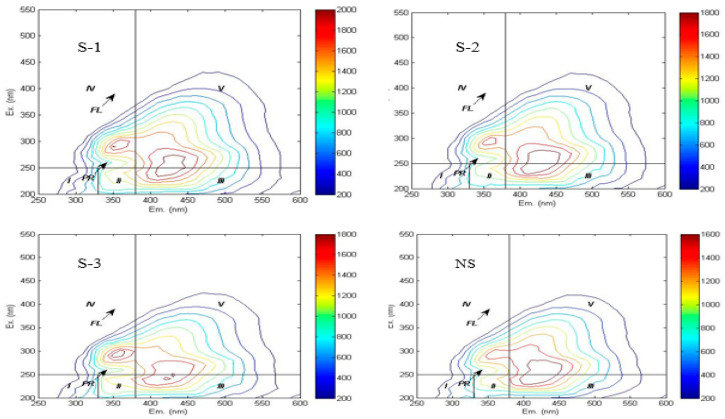
Fluorescence spectrum of different treatments.

**Figure 3 plants-15-00004-f003:**
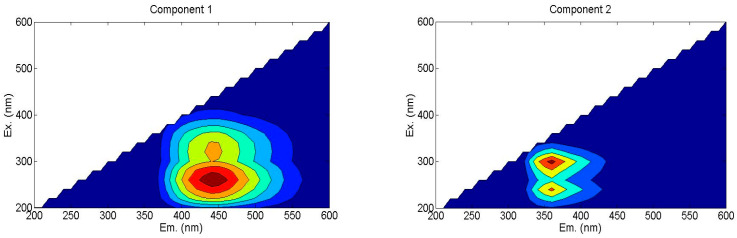
Fluorescence components of WSOC based on PARAFAC analysis method. Ex: excitation wavelength; Em: emission wavelength. C1—humic acid-like materials, C2—soluble microbial metabolites.

**Figure 4 plants-15-00004-f004:**
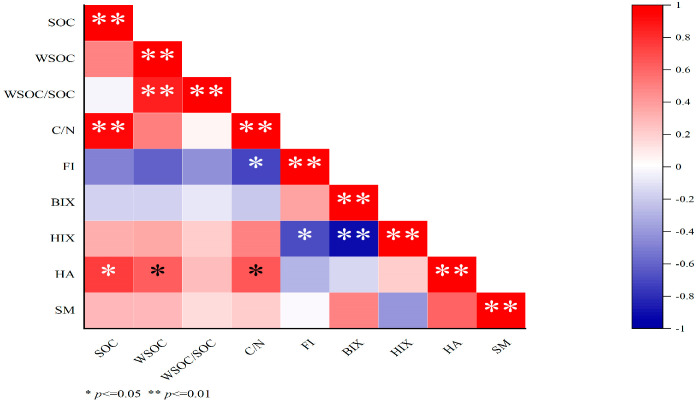
Correlation coefficients between soil WSOC and each fluorescence index. WSOC: water-soluble organic carbon; SOC: soil organic carbon; HA: humic acid-like substances; SM: soluble microbial metabolites; FI: fluorescence index; BIX: biological index; HIX: humification index.

**Table 1 plants-15-00004-t001:** Initial baseline soil chemical nutrition.

SOC g·kg^−1^	Total Nitrogen (TN) g·kg^−1^	Total Phosphorus (TP) g·kg^−1^	Total Potassium (TK)g·kg^−1^	pH (H_2_O)	Available Nitrogen (AN) mg·kg^−1^	Available Phosphorus (AP) mg·kg^−1^	Available Potassium (AK) mg·kg^−1^
18.53	2.40	2.0	21.34	6.62	103.1	70.8	167.7

**Table 2 plants-15-00004-t002:** Three-dimensional fluorescence region division.

Fluorescence Region	Types of Substances	Ex (nm)	Ex (nm)
I	Tyrosine-like protein	200–250	250–330
II	Tryptophan-like protein	200–250	330–380
III	Fulvic acid-like	200–250	380–550
IV	Soluble microbial metabolites	250–490	250–380
V	Humic-like	250–490	380–550

**Table 3 plants-15-00004-t003:** Percentage of fluorescence region integral (%).

Treatments	I Tyrosine-Like Protein	II Tryptophan-Like Protein	III Fulvic Acid-Like	IV Soluble Microbial Metabolites	V Humic Acid-Like
S-1	0.84 ± 0.12 b	7.46 ± 1.69 a	19.53 ± 1.37 a	17.86 ± 4.94 a	54.30 ± 5.16 a
S-2	0.94 ± 0.07 ab	6.50 ± 0.65 a	20.60 ± 0.80 a	14.68 ± 1.75 a	57.28 ± 1.66 a
S-3	1.02 ± 0.12 ab	6.96 ± 0.20 a	20.73 ± 0.81 a	14.93 ± 1.44 a	56.36 ± 0.73 a
NS	1.06 ± 0.03 a	6.58 ± 0.44 a	19.44 ± 0.24 a	16.09 ± 1.35 a	56.83 ± 1.56 a

Note: Different lowercase letters within columns indicate significant differences (Duncan’s test *p* < 0.05). *n* = 3.

**Table 4 plants-15-00004-t004:** Fluorescence intensity and relative percentage of soil WSOC fluorescence components.

Treatment	HA (×10^3^)	SM (×10^3^)	Total Intensity (×10^3^)	Relative Abundance (%)	
				HA	SM
S-1	1.88 ± 0.03 a	1.92 ± 0.06 a	3.8	49.53	50.47
S-2	1.73 ± 0.06 ab	1.72 ± 0.37 a	3.4	50.44	49.56
S-3	1.59 ± 0.10 bc	1.63 ± 0.34 a	3.2	49.64	50.36
NS	1.49 ± 0.10 c	1.66 ± 0.21 a	3.1	47.41	52.59

Note: Different lowercase letters within columns indicate significant differences (Duncan’s test *p* < 0.05). *n* = 3. HA: humic acid-like substances; SM: soluble microbial metabolites.

**Table 5 plants-15-00004-t005:** The mean spectral index of soil WSOC in different treatments.

Treatment	FI	BIX	HIX
S-1	1.65 ± 0.01 b	0.90 ± 0.10 a	0.64 ± 0.05 b
S-2	1.59 ± 0.02 c	0.93 ± 0.10 a	0.74 ± 0.03 a
S-3	1.61 ± 0.01 c	0.95 ± 0.07 a	0.72 ± 0.02 ab
NS	1.69 ± 0.02 a	0.95 ± 0.07 a	0.71 ± 0.02 ab

Note: The values were mean ± standard deviation, *n* = 3. Different lowercases represent a significant difference (*p* < 0.05). FI: the fluorescence index, BIX: the biological index, HIX: humification index.

**Table 6 plants-15-00004-t006:** Total straw carbon input and carbon sequestration rate.

	Total Amount of Straw Input in Five Years /kg·ha^−1^	Total Amount of Straw Carbon Input in Five Years/kg·ha^−1^	Carbon Sequestration Rate /%	Carbon Sequestration Efficiency
S-1	71,102 ± 1083	1513.2 ± 23.1	89.35	0.30
S-2	43,960 ± 816	920.8 ± 17.1	178.35	0.58
S-3	28,896 ± 392	546.7 ± 7.4	12.73	0.05
NS	-	-	-	-

## Data Availability

The data generated during and/or analyzed during the current study are available from the corresponding author. Additional information and requests for materials should be addressed to E.K.

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
