# Peer review of "Effects of Maize Straw Incorporation on Soil Water-Soluble Organic Carbon Fluorescence Characteristics"

_plants, 2025, doi:10.3390/plants15010004_

Round 1
Reviewer 1 Report (New Reviewer)
Comments and Suggestions for Authors
Dear Authors, Please find my detailed contents in the attached file.

Author Response
Title: I suggest the title as “Effects of maize straw incorporation on soil water-soluble organic carbon fluorescence characteristics”.
Response:Thank you for your suggestions. Accept.
Line 38: The word “resource” has been used two times in a single line. Try to replace/remove it.
Response:I am sorry for carelessness. We deleted it.
Lines 39 to 40: Please explain what the “other essential elements”?
Response:Thank you for your suggestions. I wanted to express “trace elements”. We changed it.
Line 42: Add the reference year in which “650 million tons” were produced.
Response:Thank you for your suggestions. Upon re-examining the straw quantity cited in the literature, I realized that the original calculation method was unclear to me. Therefore, I have now adopted the figures directly quoted from the source text. Please see below or line 40-42.
The straw production in Northeast China is 156.45 million tons, accounting for 18.1% of the total straw resources in China in 2022 [5].
Lines 45 to 52: The appropriate references are missing. Please add relevant references.
Response:Thank you for your suggestions. We added references in the manuscript.
Line 58: Please add a valid definition of “Water-soluble organic carbon” along with the reference.
Response:Thank you for your suggestions. We added the definition of WSOC in the manuscript. Please see line 57-59.
Line 61: Please add the full name of “SOC” while introducing it in the manuscript.
Response:Thank you for your suggestions. We added its full name.
Line 81: Replace “corn” with “maize”.
Response:Accept.
Lines 104 to 107: It is better if these soil analyses are presented in the form of a Table. Also, do you have these analyses for soil 20-40 cm? If yes, please add it also.
Response:Thank you for your valuable suggestion. We list all the soil analyses in the table. Sorry for not analysis the data of 20-40 cm soil layer.
Line 112: Please add the information about the agricultural machine used for crushing and deep tilling of maize straw.
Response:Thank you for your valuable suggestion. The machinery is a combined harvester by a John Deere S660(320HP) with an integrated shredding system which can complete multiple tasks such as picking maize, stacking, and stem crushing at once. The deep tilling machine is by a EurOpal -4 hydraulic flip plow (200HP), flipping the straw on the surface ground to different soil layers with soil.
Line 117: What was the tillage tool/implement used in these agronomic practices?
Response:Thank you for point out it. Agronomic practices contain field management activities, including but not limited to weeding, post-jointing cultivation of maize, top dressing, etc. the specific implementation of these practices can differ based on the practitioner. For example, management may be carried out individually by farmers or collectively at the community level. The field management is managed uniformly. Do I add this explanation in the manuscript?
Lines 141 to 148: Please add the appropriate references.
Response:Thank you for your suggestions. We added the reference.
Line 194: Figure 1 does not show the SOC. This is an important soil parameter, and I assume it was also analyzed. Moreover, it is the first part of the objective; therefore, SOC trends should be part of Figure 1, like other WSOC and WSOC/SOC.
Response:Thank you for your valuable suggestion. Accept. Please see line 188.
Line 242: Tables 3,4, and 5 need additional information about the number of replicates to get ±. And the full names of abbreviations.
Response:Thank you for your valuable suggestion. We added the full name of abbreviations.
Line 306: There is only one study [10] mentioned in the form of discussion. Please bring more relevant studies and discuss the similarities/differences of your results with others.
Response:Thank you for your valuable suggestion. Please see the revised manuscript.
Then, it will be DISCUSSION!!!!!!!!
Line 412: Table 6 needs work to be done. The column headings are the same, but the numbers below them are different. How? Please make it clear what the column headings tell us. Why do both have the same headings but different numbers?
Response:Thanks again for your patience and sharp eyes. This was my mistake from copying, and I missed it in the final check before submission. I marked the right title in Table 6.
Line 440 to 448: “Herein, the main objectives of this study were to (1) determine whether continuous straw returning increases the instability of the soil carbon pool”. Is the soil carbon pool referred to as SOC? If yes, then this part of the objective is not answered in the conclusions
Response:Thanks again for your valuable suggestion. We revised in the manuscript. Please see below or line 455-459.
Reviewer 2 Report (New Reviewer)
Comments and Suggestions for Authors
The article “Effects of maize straw incorporation on soil water soluble organic carbon fluorescence characteristics in crop fields” provides detailed information on the effects of maize straw incorporation on the distribution of soluble carbon in different soil layers. The presented studies are relevant for providing soil with organic nutrients and from an environmental point of view.
Observations:
- In subsection “2.1. Site description” (line 98) it is written that the studies were carried out in 2011-2015, the question arises why the studies were not continued, why are ten-year-old results presented?
- I suggest presenting Figures 1, 2 and 4 more clearly (font size, brightness).
- In subsection “2.3. Measurement Methods” (line 124) it should be explained and better justified why the study samples were taken in the 0-20 cm and 20-40 cm layers.
- I suggest that the amounts of incorporated straw be presented in kg ha-1. It should also be explained how much straw can be incorporated to achieve better yield results.
- Lines 110-112 state “The trial employed maize (Zea mays L.) straw as incorporation material, with the following experimental components: Post-harvest maize straw (average yield: 10000 kg·hm-2) was mechanically crushed (<5 cm fragments) and deep-tilled (30-35 cm)”, which raises the question of why the tests were carried out at a depth of 0-20 cm?
- The conclusion states “Furthermore, the biennial straw return mode was identified as optimal for stabilizing soil organic carbon.” (lines 446 - 447), it would be good to indicate how much straw is recommended to be incorporated?
- If possible, I suggest providing more recent references 1, 2 and 3.
Author Response
Observations:
- In subsection “2.1. Site description” (line 98) it is written that the studies were carried out in 2011-2015, the question arises why the studies were not continued, why are ten-year-old results presented?
Response:Thank you for the sharp question. The data source provided this information to me in 2016. After a previous draft was rejected many times and continuously improved manuscript. Given the substantial work invested in this part, I’ve decided to publish it. Follow-up data is still being processed.
- I suggest presenting Figures 1, 2 and 4 more clearly (font size, brightness).
Response:Thank you for your valuable suggestion. I will afford independent images when submission.
- In subsection “2.3. Measurement Methods” (line 124) it should be explained and better justified why the study samples were taken in the 0-20 cm and 20-40 cm layers.
Response:Thank you for your valuable suggestion. Subsoil nutrient content is increasingly acknowledged as vital for crop growth. We focused on the 0-35 cm layer-the depth for straw incorporation via deep plowing- to measure the increase in subsoil organic carbon and WSOC.
- I suggest that the amounts of incorporated straw be presented in kg ha-1. It should also be explained how much straw can be incorporated to achieve better yield results.
Response:Thank you for your valuable suggestion. Actually, you are right, we revised this part in Table 6.
- Lines 110-112 state “The trial employed maize (Zea mays L.) straw as incorporation material, with the following experimental components: Post-harvest maize straw (average yield: 10000 kg·hm-2) was mechanically crushed (<5 cm fragments) and deep-tilled (30-35 cm)”, which raises the question of why the tests were carried out at a depth of 0-20 cm?
Response:Topsoil is vital for crop growth, which is why many researchers, myself included, examine soil nutrients and structure across different layers. A common approach is to divide the soil into 0–15 cm and 15–30 cm layers, a division aligned with typical tillage depths. I would like to note that the collection of these soil samples took place earlier, and such specific stratification was not initially planned.
- The conclusion states “Furthermore, the biennial straw return mode was identified as optimal for stabilizing soil organic carbon.” (lines 446 - 447), it would be good to indicate how much straw is recommended to be incorporated?
Response:This article is premised on the integrated utilization of crop straw. It aims to provide insights into this field by arguing that while continuous annual straw incorporation increases soil organic carbon, the newly added straw carbon can also prime the decomposition of native soil carbon, potentially undermining its long-term sequestration. Conversely, biennial straw return also can increase SOC. This practice liberates straw in alternate years for other purposes, thus opening up an additional avenue for its comprehensive use.
- If possible, I suggest providing more recent references 1, 2 and 3.
Response:Thank you for your valuable suggestion. Accept.

Reviewer 3 Report (New Reviewer)
Comments and Suggestions for Authors
The authors have made the suggested corrections.
The manuscript demonstrates quality and scientific merit for publication.
The only observation is that, although the authors have made the corrections, these corrections are not easily identifiable. Files should be created to clearly indicate the corrections suggested by the reviewers.
The corrections and/or suggestions were made by the authors.
- What is the main question addressed by the research?
Effects of maize straw incorporation on soil water soluble organic carbon fluorescence characteristics in crop fields
- Do you consider the topic original or relevant to the field? Does it address a specific gap in the field? Please also explain why this is/ is not the case.
Yes, the investigation seeks to elucidate information on the impacts of long-term straw incorporation frequencies—including annual (S-1), biennial (S-2), and triennial (S-3) return patterns—on WSOC distribution across 0-20 cm and 20-40 cm soil profiles.
- What does it add to the subject area compared with other published material?
The importance of the research is referenced by the research methodologies used by the authors, through the integration of three-dimensional excitation-emission matrix (EEM) fluorescence spectroscopy with parallel factor analysis (PARAFAC)
- What specific improvements should the authors consider regarding the methodology? What further controls should be considered?
The methodology is good, no suggestions.
- Are the conclusions consistent with the evidence and arguments presented and do they address the main question posed? Please also explain why this is/is not the case.
The conclusion is good, no suggestions.
- Are the references appropriate?
Yes.
- Any additional comments on the tables and figures.
There is no recommendation.
Author Response
The manuscript demonstrates quality and scientific merit for publication.
The only observation is that, although the authors have made the corrections, these corrections are not easily identifiable. Files should be created to clearly indicate the corrections suggested by the reviewers.
Response:Thank you for your suggestions. We revised a lot in the revised manuscript. If you have other questions, please let me know.
The corrections and/or suggestions were made by the authors.
- What is the main question addressed by the research?
Effects of maize straw incorporation on soil water soluble organic carbon fluorescence characteristics in crop fields
- Do you consider the topic original or relevant to the field? Does it address a specific gap in the field? Please also explain why this is/ is not the case.
Yes, the investigation seeks to elucidate information on the impacts of long-term straw incorporation frequencies—including annual (S-1), biennial (S-2), and triennial (S-3) return patterns—on WSOC distribution across 0-20 cm and 20-40 cm soil profiles.
- What does it add to the subject area compared with other published material?
The importance of the research is referenced by the research methodologies used by the authors, through the integration of three-dimensional excitation-emission matrix (EEM) fluorescence spectroscopy with parallel factor analysis (PARAFAC)
- What specific improvements should the authors consider regarding the methodology? What further controls should be considered?
The methodology is good, no suggestions.
- Are the conclusions consistent with the evidence and arguments presented and do they address the main question posed? Please also explain why this is/is not the case.
The conclusion is good, no suggestions.
- Are the references appropriate?
Yes.
- Any additional comments on the tables and figures.
There is no recommendation.

Reviewer 4 Report (New Reviewer)
Comments and Suggestions for Authors
Dear All,
This manuscript explores the influence of different frequencies of maize straw incorporation (annual, biennial, triennial) on the content and composition of water-soluble organic carbon (WSOC) and its fluorescence characteristics in black soils of Northeast China. The study applies three-dimensional excitation-emission matrix (EEM) fluorescence spectroscopy and PARAFAC modeling to assess the chemical and structural properties of WSOC components.
The experimental approach is sound, and the analytical methodology is advanced. However, there are critical omissions in reporting soil edaphic conditions, particularly soil physical attributes, soil taxonomy, and classification, which are essential for reproducibility, ecological interpretation, and broader applicability of the findings.
* Strong Points *
-
The manuscript uses high-resolution optical methods (EEM-PARAFAC) to decompose WSOC into humic acid-like and microbial metabolite-like components.
-
The fluorescence indices (FI, BIX, HIX) are well-calculated and interpreted in relation to straw return regimes.
-
Field-based experimental design with replication across multiple years adds robustness.
-
The study's applied relevance for carbon stabilization and straw resource management is timely and well-integrated into the discussion.
* Weaker Aspects and Required Revisions *
1. Lack of Soil Taxonomy and Classification
Despite clear description of chemical parameters (Page 3, Lines 103–107), the soil type is only broadly referred to as “black soil”, without providing:
-
FAO classification (e.g., Chernozem, Phaeozem)
-
Chinese soil taxonomy equivalent
-
Texture class (e.g., sandy loam, clay loam)
-
Particle size distribution or bulk density
* Reviewer comment: Please provide official soil classification according to FAO/WRB, USDA, or Chinese Soil Taxonomy. Additionally, include soil texture and bulk density. These are standard descriptors and crucial for understanding carbon retention, water dynamics, and microbial activity.
2. Incomplete Description of Soil Physical Attributes
Only chemical parameters are listed in Line 103–107. Missing:
-
Sand, silt, and clay fractions
-
Water holding capacity
-
Soil structure (granular, blocky, etc.)
-
Depth to hardpan or compaction layer (if relevant)
* Reviewer comment: Soil physical properties should be fully described (e.g., texture, porosity, compaction risks), especially as deep tillage is central to straw incorporation. Physical attributes influence SOC mobility and fluorescence signal interpretation.
3. Climatic Conditions – Acceptable but Could Be Strengthened
Climatic conditions are outlined clearly (Page 3, Lines 99–102):
-
Annual mean temp: 5.2°C
-
Accumulated temp ≥10°C: 2,600–3,000°C
-
Rainfall: 450–650 mm (70% in summer)
-
Location: Gongzhuling, Jilin Province
* Reviewer suggestion: It would be valuable to indicate soil moisture regime (e.g., Udic, Ustic) and growing season length. Also, note whether waterlogging or drought events occurred during the trial years (2011–2016), as these could influence WSOC fluorescence.
4. Clarification of Straw Quantity and Frequency
-
It is unclear if equal straw amounts were returned across treatments or if the frequency influenced total input.
-
Table 6 (Page 11) suggests S-2 applied less straw overall but had higher carbon sequestration efficiency, which is counterintuitive and underexplained.
* Reviewer suggestion: Please clarify whether straw inputs were normalized across treatments (e.g., S-1 applied 10 t/ha/year vs. S-2 20 t/ha every 2 years), and indicate C:N ratio of the applied straw.
5. Spectral Interpretation Needs Deeper Mechanistic Context
While the spectral zones and PARAFAC outputs are well-structured, the biological implications of:
-
blue/red shifts,
-
increased humic acid-like compounds (C1),
-
decline in microbial metabolites (C2),
are insufficiently mechanistically explained.
* Reviewer suggestion: Enhance discussion on microbial processing pathways, humification chemistry, and molecular weight dynamics, linking back to FI, BIX, HIX results.
* Language and Style Issues *
The manuscript is largely intelligible, but several sentences suffer from unclear syntax or repetition. Examples:
-
Page 2, Line 72–74: "This study aims to make sure the specific changes..." → Suggest: "This study aims to characterize changes in WSOC content and composition induced by straw incorporation."
-
Page 3, Line 99: “pre-experimental soil analysis” → Suggest: “Initial baseline soil characterization”
-
Page 11, Line 386: “While there was no straw decomposition to compete…” is unclear — the phrase needs revision.
* Reviewer recommendation: I recommend minor language polishing by a professional editor, particularly in the Introduction and Discussion.
Author Response
- Lack of Soil Taxonomy and Classification
Despite clear description of chemical parameters (Page 3, Lines 103–107), the soil type is only broadly referred to as “black soil”, without providing:
- FAO classification (e.g., Chernozem, Phaeozem)
- Chinese soil taxonomy equivalent
- Texture class (e.g., sandy loam, clay loam)
- Particle size distribution or bulk density
* Reviewer comment: Please provide official soil classification according to FAO/WRB, USDA, or Chinese Soil Taxonomy. Additionally, include soil texture and bulk density. These are standard descriptors and crucial for understanding carbon retention, water dynamics, and microbial activity.
Response:Thank you for your valuable suggestion. We revised in the manuscript. Please see as below.
Soil was a typical black soil (aligned with Phaeozem in the Chinese soil taxonomy) developed from loess parent material which particle size distribution was sand of 38.3%, silt 29.9% and clay of 31.8%.
- Incomplete Description of Soil Physical Attributes
Only chemical parameters are listed in Line 103–107. Missing:
- Sand, silt, and clay fractions
- Water holding capacity
- Soil structure (granular, blocky, etc.)
- Depth to hardpan or compaction layer (if relevant)
* Reviewer comment: Soil physical properties should be fully described (e.g., texture, porosity, compaction risks), especially as deep tillage is central to straw incorporation. Physical attributes influence SOC mobility and fluorescence signal interpretation.
Response:Thank you for your valuable suggestion. We revised in the manuscript. Please see as below.
Soil was a typical black soil (aligned with Phaeozem in the Chinese soil taxonomy) developed from loess parent material which particle size distribution was sand of 38.3%, silt 29.9% and clay of 31.8%.
But for other soil physical characteristics were missed. Soil sampling occurred after the field had been plowed, so no physical properties were measured for the initial soil character. Due to the consistent use of deep plowing for all annual treatments, soil bulk density and field capacity were managed uniformly across the experimental area.
- Climatic Conditions – Acceptable but Could Be Strengthened
Climatic conditions are outlined clearly (Page 3, Lines 99–102):
- Annual mean temp: 5.2°C
- Accumulated temp ≥10°C: 2,600–3,000°C
- Rainfall: 450–650 mm (70% in summer)
- Location: Gongzhuling, Jilin Province
* Reviewer suggestion: It would be valuable to indicate soil moisture regime (e.g., Udic, Ustic) and growing season length. Also, note whether waterlogging or drought events occurred during the trial years (2011–2016), as these could influence WSOC fluorescence.
Response:Thank you for your valuable and important suggestion. I’ve checked some records, and it appears that no severe weather occurred during the trial. I will make a point to track and record weather conditions going forward.
- Clarification of Straw Quantity and Frequency
- It is unclear if equal straw amounts were returned across treatments or if the frequency influenced total input.
- Table 6 (Page 11) suggests S-2 applied less straw overall but had higher carbon sequestration efficiency, which is counterintuitive and underexplained.
* Reviewer suggestion: Please clarify whether straw inputs were normalized across treatments (e.g., S-1 applied 10 t/ha/year vs. S-2 20 t/ha every 2 years), and indicate C:N ratio of the applied straw.
Response:Thank you for your valuable suggestion. Under this scheme, the annual return treatment applies the full straw amount (~10,000 kg·ha⁻¹) each year, while the biennial and triennial treatments receive reduced total inputs. A detailed layout diagram was originally included but was omitted following a previous reviewer's suggestion. Consequently, in terms of total straw applied, the S- treatment (annual) involves the highest quantity.
|
Treatment |
2011 |
2012 |
2013 |
2014 |
2015 |
The total amount of straw |
|
S-1 |
Yes |
Yes |
Yes |
Yes |
Yes |
5×10000 kg·ha-1 |
|
S-2 |
Yes |
No |
Yes |
No |
Yes |
3×10000 kg·ha-1 |
|
S-3 |
Yes |
No |
No |
Yes |
No |
2×10000 kg·ha-1 |
|
NS |
No |
No |
No |
No |
No |
0 |
- Spectral Interpretation Needs Deeper Mechanistic Context
While the spectral zones and PARAFAC outputs are well-structured, the biological implications of:
- blue/red shifts,
- increased humic acid-like compounds (C1),
- decline in microbial metabolites (C2),
are insufficiently mechanistically explained.
* Reviewer suggestion: Enhance discussion on microbial processing pathways, humification chemistry, and molecular weight dynamics, linking back to FI, BIX, HIX results.
Response:Thank you for your valuable suggestion. We revise in the manuscript. Please see line 366-373.
* Language and Style Issues *
The manuscript is largely intelligible, but several sentences suffer from unclear syntax or repetition. Examples:
- Page 2, Line 72–74: "This study aims to make sure the specific changes..." → Suggest: "This study aims to characterize changes in WSOC content and composition induced by straw incorporation."
- Page 3, Line 99: “pre-experimental soil analysis” → Suggest: “Initial baseline soil characterization”
- Page 11, Line 386: “While there was no straw decomposition to compete…” is unclear — the phrase needs revision.
Response:Thank you for your suggestions. Accept all. The third suggestion we deleted the sentence “While there was no straw decomposition to compete for energy, the scarcity of available carbon ultimately promotes microbial growth.”
* Reviewer recommendation: I recommend minor language polishing by a professional editor, particularly in the Introduction and Discussion.
Response:Thank you for your valuable suggestion. I revised the manuscript carefully.

Round 2
Reviewer 1 Report (New Reviewer)
Comments and Suggestions for Authors
The recommendations have been incorporated therefore, the manuscript can be accepected in the present form.
Author Response
Thank you for your suggestions. All the revised in the manuscript.
Reviewer 4 Report (New Reviewer)
Comments and Suggestions for Authors
Dear All,
Thank you for your thoughtful and detailed revisions. I have carefully reviewed the revised manuscript entitled: "Effects of maize straw incorporation on soil water-soluble organic carbon fluorescence characteristics in crop fields" (plants-4022513).
The manuscript has been substantially improved in structure, clarity, and overall presentation. Most of the concerns raised during the previous review have been adequately addressed. The experimental design is solid, the analytical methods are appropriate, and the use of EEM-PARAFAC adds valuable insight into WSOC characterization under different straw return regimes.
That said, a few important aspects still require minor clarification to ensure the study’s conclusions are fully supported and clearly communicated.
* Positive Aspects
-
The objectives are now better defined, and the methodology, especially regarding PARAFAC modeling and spectral index calculation, is clearly described.
-
Figures and tables are well integrated into the narrative, and the statistical comparisons are clearly presented.
-
The authors now provide a clearer explanation of the fluorescence indices and their general ecological relevance.
-
The discussion has improved in tone and interpretation, with fewer overstatements and better linkage to existing literature.
* Points That Still Require Attention
-
Hypothesis Framing
The introduction still presents an expected outcome ("more straw = more WSOC in surface soils") rather than a testable or novel hypothesis. I suggest reframing the objective to emphasize the qualitative changes in WSOC composition and humification, rather than only its concentration. -
Vertical WSOC Distribution
The manuscript interprets depth-dependent WSOC trends as treatment effects. However, it does not consider the influence of deep tillage, which could have redistributed organic matter independently of straw incorporation. Please acknowledge this potential confounding factor in the discussion. -
Interpretation of Fluorescence Indices
The mechanistic meaning of FI, BIX, and HIX is still somewhat generic. Briefly contextualize these indices in terms of microbial activity, decomposition stage, or aromaticity, especially in relation to PARAFAC components C1 and C2. -
Ecological Meaning of C1/C2
The manuscript would benefit from a deeper discussion of how the two PARAFAC components (humic-like vs. microbial metabolites) reflect soil carbon cycling or microbial turnover, especially under different straw return frequencies. -
Language Polishing
While greatly improved, some sentences, particularly in the Discussion, still require refinement for clarity and tone. A final round of language editing is recommended.
Author Response
- Hypothesis Framing
The introduction still presents an expected outcome ("more straw = more WSOC in surface soils") rather than a testable or novel hypothesis. I suggest reframing the objective to emphasize the qualitative changes in WSOC composition and humification, rather than only its concentration.
Response: Thank you for your valuable suggestion. Please see below or line 84-86, marking red. We hypothesize the practice of maize straw application influences WSOC fractions of fluorescent materials. Herein, the main objectives of this study were to (1) determine whether annual straw returning increase the instability of soil carbon pool and (2) annual straw return to the field will reduce the stable components of WSOC and decrease humification. This study aims to characterize changes in WSOC content and composition induced by straw incorporation.
- Vertical WSOC Distribution
The manuscript interprets depth-dependent WSOC trends as treatment effects. However, it does not consider the influence of deep tillage, which could have redistributed organic matter independently of straw incorporation. Please acknowledge this potential confounding factor in the discussion.
Response: Thanks again for your important suggestion. We revised in the manuscript and see line 307-318, highlighted in red.
- Interpretation of Fluorescence Indices
The mechanistic meaning of FI, BIX, and HIX is still somewhat generic. Briefly contextualize these indices in terms of microbial activity, decomposition stage, or aromaticity, especially in relation to PARAFAC components C1 and C2.
Response: This is a very sharp and important question. Thank you very much point out it. We added this part in the manuscript. please see the line 390-412, highlighted in red.
- Ecological Meaning of C1/C2
The manuscript would benefit from a deeper discussion of how the two PARAFAC components (humic-like vs. microbial metabolites) reflect soil carbon cycling or microbial turnover, especially under different straw return frequencies.
Response: Thank you for your excellent question, which is crucial for clarifying the research context. However, I must apologize-the time originally allotted for revising the manuscript was quite limited, and I am still a new one in fluorescence spectrum. To better understand and refine the content, I referred to additional literature. Recently, though, I have noticed considerable inconsistencies and even contradictions among different resources. Tracing back the references has unfortunately added to the confusion rather than resolving it, which has slowed my progress in revising the manuscript. therefore, I kindly request additional time to thoroughly review the literature and improve the manuscript. If you have and directly relevant and authoritative references on hand, would you be willing to share them with me? This would help me cite more accurately and address your points more effectively.
And then, I received a reminding email today, which made me feel anxious. I will submit the manuscript first and look forward to your response. Thank you so much for the thoughtful and detailed questions you raised regarding my manuscript. Thank you again.
- Language Polishing
While greatly improved, some sentences, particularly in the Discussion, still require refinement for clarity and tone. A final round of language editing is recommended.
Response: Accept.
This manuscript is a resubmission of an earlier submission. The following is a list of the peer review reports and author responses from that submission.
Round 1
Reviewer 1 Report
Comments and Suggestions for Authors
It was hypothesized the application of maize straw obtained at three different return years varied WSOC fractions of fluorescent materials. Herein, the main objectives of this study were to (1) WSOC concentrations had a higher content in continuous(S-1) treatment; (2) triennial(S-3) and no straw incorporation (NS) treatments had little changes on WSOC fractions because of poor straw input. the Results of the studies confirmed that the characteristics of various fluorescent components of WSOC and provide theoretical sup- port for the diversified utilization of straw in Northeast China
- Do you consider the topic original or relevant to the field? Does it address a specific gap in the field? Please also explain why this is/ is not the case.
Yes, the topic is original, therefore this study systematically investigated the impacts of long-term straw incorporation frequencies-including continuous(S-1), biennial(S-2), and triennial(S-3) return patterns-on WSOC distribution across 0-20 cm and 20-40 cm soil profiles. Through the integration of three-dimensional excitation-emission matrix (EEM) fluorescence spectroscopy with parallel factor analysis (PARAFAC), it was elucidated structural characteristics and humification dynamics associated with different incorporation regimes.
- What does it add to the subject area compared with other published material?
Potential to improve the fertility of weaker soils, which is important from an environmental, and the possibilities of straw management and crop yield perspective.
- What specific improvements should the authors consider regarding the methodology?
I have no comment. Is ok.
- Are the conclusions consistent with the evidence and arguments presented and do they address the main question posed? Please also explain why this is/is not the case.
Yes, I have no comments
- Are the references appropriate?
Yes, but the following paragraphs should be added to the discussion with a review of the comments.
Discussion
After continuous straw returning (S-1 treat-242 ment), tryptophan-like protein substances and soluble microbial metabolites increased, 243 tyrosine-like protein substances and humus-like substances decreased, and fulvic acid-244 like substances had no obvious change trend in these treatments (position of literature..) .After continuous straw returning (S-1 treat-242 ment), tryptophan-like protein substances and soluble microbial metabolites increased, 243 tyrosine-like protein substances and humus-like substances decreased, and fulvic acid-244 like substances had no obvious change trend in these treatments. Studies have shown that 245 protein-like components were an important factor to control the characteristic fluctuation 246 of dissolved organic matter. There was a significant difference in the proportion of pro-247 tein-like/humic-like fluorescent components in soil dissolved organic matter after land 248 tillage change (posiotion of literature). In this experiment, the humification degree of dissolved organic carbon 249 among all treatments was the lowest with the continuous straw return, showing a weak 250 humification degree. There was no significant difference among other treatments (posiotion of literature) . The 251 sources of dissolved organic matter were the dual contributions of internal (microorgan-252 isms, algae) and external (humus) [32].Studies have shown that 245 protein-like components were an important factor to control the characteristic fluctuation 246 of dissolved organic matter. There was a significant difference in the proportion of pro-247 tein-like/humic-like fluorescent components in soil dissolved organic matter after land 248 tillage change. In this experiment, the humification degree of dissolved organic carbon 249 among all treatments was the lowest with the continuous straw return, showing a weak 250 humification degree. There was no significant difference among other treatments. The 251 sources of dissolved organic matter were the dual contributions of internal (microorgan-252 isms, algae) and external (humus) [32].
- Any additional comments on the tables and figures . I have no comment.
Best regards
Reviewer 2 Report
Comments and Suggestions for Authors
This manuscript presents a comprehensive study exploring how the frequency of maize straw incorporation into agricultural soil affects the distribution and quality of water-soluble organic carbon (WSOC), as well as its fluorescence characteristics. The study is timely, relevant, and grounded in a well-structured long-term field experiment. It uses advanced spectroscopic techniques, notably three-dimensional excitation-emission matrix (EEM) fluorescence combined with parallel factor analysis (PARAFAC), to dissect the chemical nature of dissolved organic matter (DOM) in soils subject to various straw management regimes.
Overall, the paper addresses a significant research gap regarding the temporal dynamics of WSOC quality under different straw incorporation schedules. The authors should be commended for the depth of experimental design and the diversity of analytical techniques employed. However, several elements of the manuscript—especially in the introduction, discussion, and the overall narrative cohesion—require substantial clarification and revision before it can be considered suitable for publication.
In the introduction (lines 37–74), the authors present relevant background information and cite key references related to the impacts of straw return on soil organic matter and nutrient dynamics. However, the current framing lacks precision in identifying the specific research gap. The authors mention the application of EEM and fluorescence spectroscopy in prior studies but do not fully articulate what is novel in their own approach. Moreover, the hypothesis outlined in lines 68–73 is not clearly formulated; the structure of the sentence is grammatically awkward and lacks clarity. A more precise statement of the hypothesis—linked to the expected differences in WSOC fractions under different straw regimes—is needed.
The results section (lines 75–179) is generally well organized, but at times, the interpretation of findings could be more robust. For example, in lines 82–87, while it is stated that continuous straw return (S-1) increased WSOC content and WSOC/SOC ratios in the topsoil, there is no accompanying explanation of the implications of this observation for soil quality or carbon stabilization. Likewise, Figure 1 and Table 2 are informative, but they are not fully integrated into the narrative. The authors should aim to better contextualize these findings with reference to soil carbon turnover and microbial accessibility.
A particular strength of the manuscript lies in its detailed spectral analysis of WSOC components (lines 93–145). The fluorescence peak assignments and PARAFAC component interpretations are clearly presented, and the spectral shifts observed across treatments are intriguing. However, the discussion of these shifts (lines 111–116) remains superficial. The authors mention “blue shift” and “red shift” phenomena, but the biochemical implications of these spectral changes—especially in relation to molecular size, aromaticity, or microbial processing—are not adequately elaborated. This section would benefit from a more mechanistic interpretation supported by relevant literature.
The discussion section (lines 180–303), though rich in references, would greatly benefit from more coherent structuring and critical synthesis. Presently, the discussion oscillates between restating results and introducing new information without clearly demarcated themes. For example, from lines 180 to 232, the discussion spans multiple topics—soil depth gradients, straw-induced microbial activity, and carbon turnover—without sufficient linkage or thematic transitions. Furthermore, the claims made in lines 292–297 regarding the degradation of fulvic substances following continuous straw return are intriguing but insufficiently substantiated. It would strengthen the manuscript to explicitly cite studies that report on microbial decomposition of fulvic acid-like compounds and link those processes to observable fluorescence indices (FI, BIX, HIX).
A further concern is the explanation of the PARAFAC-derived components and their ecological interpretation (lines 147–153). While the statistical outputs are sound, the ecological roles of C1, C2, and C3 components remain underdeveloped. What does a higher relative abundance of tryptophan-like substances, for instance, suggest about the microbial activity or DOM lability in these soils? These aspects should be explicitly discussed to ensure that the fluorescence data are not just descriptive, but also explanatory.
Methodologically, the study is solid. The experimental design, as outlined in lines 304–329, reflects sound agricultural practice and adequate replication. The fluorescence spectroscopy and data processing steps are described with sufficient detail. However, minor clarifications are needed: for example, line 356 mentions correction for Raman scattering during PARAFAC processing—this should include a brief note on how this was achieved (e.g., subtraction, masking, or mathematical correction) since it can influence component resolution.
The conclusions (lines 364–375) summarize the study’s main findings but would benefit from greater emphasis on practical implications. While the authors suggest that biennial incorporation offers an optimal balance between stabilization and resource allocation, they do not specify how these findings might influence field-level management decisions. Additionally, the link between observed spectral properties and long-term soil health remains somewhat indirect; this connection could be more clearly drawn.
The manuscript’s language is largely intelligible, though not fluent throughout. There are noticeable grammatical and syntactic errors, especially in complex technical sentences (e.g., lines 66–73, 180–190, 253–259). Professional language editing is strongly recommended to improve clarity, reduce redundancy, and enhance scientific tone.
In summary, this manuscript has considerable potential. It offers important insights into the biochemical behavior of WSOC under straw amendment regimes using advanced analytical techniques. However, several areas need to be revised or expanded: the hypothesis must be better defined; spectral interpretations should be enriched with mechanistic discussion; and the overall clarity of language requires improvement. With these revisions, the manuscript will make a valuable contribution to the literature on organic carbon dynamics in managed agroecosystems.
Recommendation: Major revisions required.
Reviewer 3 Report
Comments and Suggestions for Authors
General comments:
This manuscript proposes a hypothesis that is very weak (without scientific novelty): if straw is applied more often it will provide more water-soluble organic carbon, and that the more organic carbon will be found in the soil surface than in deep soil layers. Following this hypothesis, this article has little significance. However, it has interesting results that require a different approach. The interesting content of this article is the capacity of the method to measure several fractions of the organic carbon, and explore how those fractions were influenced by the treatments. My suggestion is that the manuscript is reformulated with little emphasis on the hypothesis commented above, but proposing a more innovative hypothesis regarding the fractions of organic matter, the indexes and other interesting information that can be extracted from these results.
When presenting and discussing the results, the authors should not disregard that the straw was incorporated into the soil (with deep-tillering), and this procedure strongly influence how the organic carbon was distributed in the soil profile. It is likely that most of the difference between the two soil layers was caused by the tillering rather than a regular dynamics of organic carbon in the soil.
Abstract
Line 17: the term continuous suggest to the reader a more frequent straw deposition (like weekly). Consider calling the treatment as “annual”.
Line 25: Please avoid abbreviations when possible because it is difficult to the reader - replace “S-2 treatment” with “Biennial return”.
Introduction
The introduction is not organized as a sequence of ideas that leads to the knowledge gap or the hypothesis of this study. It is just a compilation of phrases informing some previous research, lacking a link among them, and without a clear purpose.
This section needs to be rewritten with the focus on presenting the background knowledge on the subject, emphasizing the knowledge gap that this study proposes to fill, and stating clearly what are the hypotheses or objectives of the study. Please break the Introduction in paragraphs according to the subject being discussed. Allow a separate paragraph for the objective/hypothesis (lines 68-74).
Line 40: Please clarify what is “deep field”. Do you mean “deep soil layers”?
Line 43: Please inform briefly or mention some of the “the ways of straw incorporation”.
Line 44: It is not clear what are the alternatives which “either way” refers to.
Line 45: Do you mean “energy use efficiency” instead of “energy reuse”?
Libe 49: Delete “at home and abroad”.
Lines 51-55: Please revise these lines. It is not clear what “has an obvious effect on the fluorescence peaks”, and I confused if “they (they what?) affect tillage” or if “they are affected by tillage…” The main idea of these sentences should be rephrased.
Lines 68-69: Suggestion: “We hypothesize the patter of maize straw application influences WSOC fractions…”
Lines 69-71: It seems to me that these lines could be deleted, but if the authors keep them, please revise the grammar and rephrase it clearly. Maybe you mean that the WSOC will be higher if straw is applied every year instead of every three years (this objective seems obvious – please consider if it should be stated).
Lines 72-74: Please delete these lines. This could be considered later as part of the discussion, not in the Objectives.
Results
Line 81: “(20-40 cm) compared with NS,…”
Line 86: consider replacing “surface horizons” with “soil surface” or clarify what you mean with those terms.
Figure 1: You would emphasize the contrast among treatment changing the scale of the vertical axis to 200 to 400 mg/kg (graph on the left) and 1.2 to 1.8 % (graph on the right).
Line 116: Can you clarify what is “red shift”?
Table 4: Write in full FI, BIX, and HIX; in the footnote, there is not “very significant”, it is either significant or not-significant.
Table 5: R2 is not an appropriate information for this analysis, and it is presented as p-value (that is what the title informs). Consider deleting the two bottom rows of the table.
Discussion
Lines 192-199: This is a poor explanation on the organic matter decrease in the soil profile. It is questionable if clay content is the driver of this effect, while is obvious that as soil surface has more organic carbon because straw is applied on the soil surface. The reasoning of the microbial decomposition is the opposite of what was proposed – the reduced decomposition would lead to more organic carbon, not to less OC. Please consider if the term “dual role” for the straw could be replaced with “dual effect”.
Lines 200-212: I am not convinced by the reasons and mechanisms proposed to explain the slight changes on WSOC ratios observed in the study. It seems that the authors are extrapolating trying to provide an explanation to an effect that is closer to occasional than to a scientifically-sound observation (specially to find some explanation to S2 lying between S1 and S3). In a five-years experiment, it should not be considered relevant or to make reasonable inferences on the mechanisms driving the changes on soil organic carbon and their contrast between soil layers. It seems that the results from this study does not allow such a complex elucidation, and the discussion should be more restricted to the results.
It seems that this discussion was written for a different paper. In line 220, what you mean with “straw removal” (did it happen?) and how would it promote OC formation? In line 248, the text is discussing “land tillage change”, but this manuscript is on a different subject.
Lines 301-303: I question if the humic compounds can really be converted in “fulvic acid-like components”, but the most important question is how this conversion will “improve the soil's nutrient supply capacity”, as stated. Please consider it critically.
Material and methods
Please clarify if the amount of straw applied was equal in each application or in the sum of all years. For instance, in S-3, was the amount 3 t/ha or 1 t/ha? In S-2, was each application made with 2 t/ha or 1 t/ha?
The statistical procedures were not properly described. More details should be provided on the calculation of the three indexes (Fluorescence Index, Biological Index and Humification Index). And the meaning of these indexes should be further explained in the Introduction (or when they are discussed). The multivariate analysis were not explained (what are the variables, how were they analyzed, what is the method?)
Line 306: Do you mean a “long-term experiment” (what makes sense) instead of a “long-term experimental station” (what is the point with the duration of the experimental station?). Is this a long-term experiment (like 30 years long) or simply a five-year experiment as described ahead?
Line 313-318: Avoid the “respectively” (it is not necessary, and it is very difficult to the reader). It is easier writing “18.53 g·kg-1 of soil organic carbon, 2.40 g·kg-1 of total nitrogen…”
Line 327: “Agronomic practices were employed equally to all the plots, such as fertilization…”
Table 6 is not necessary and could be deleted. The only information needed is the clarification that for S-2, straw was applied 3 times (including the last year), and that for S-3, the straw was applied on year 1 and 4.
Conclusions
Lines 373-375: I suggest to consider critically if the results from this study are strong enough to support the conclusion that returning straw every two years can sustain better soil properties than annual straw application. This conclusion is highly questionable and difficult to support.
Comments on the Quality of English LanguageEnglish usage
The manuscript requires a thorough review. There are many lacking verbs, missing commas, scientific names without italics, and many other typos that were not commented.